# Enhancing Human Activity Recognition in Smart Homes with Self-Supervised Learning and Self-Attention [note 1]

**DOI:** 10.3390/s24030884

**Published:** 2024-01-29

**Authors:** Hui Chen, Charles Gouin-Vallerand, Kévin Bouchard, Sébastien Gaboury, Mélanie Couture, Nathalie Bier, Sylvain Giroux

**Affiliations:** 1Department of Computer Science, Université de Sherbrooke, 2500 Bd de l’Université, Sherbrooke, QC J1K 2R1, Canada; hui.chen@usherbrooke.ca (H.C.); sylvain.giroux@usherbrooke.ca (S.G.); 2Department of Computer Science and Mathematics, Université du Québec à Chicoutimi, 555 Bd de l’Université, Chicoutimi, QC G7H 2B1, Canada; kevin_bouchard@uqac.ca (K.B.); sebastien_gaboury@uqac.ca (S.G.); 3Faculty of Arts and Humanities, Université de Sherbrooke, 2500 Bd de l’Université, Sherbrooke, QC J1K 2R1, Canada; melanie.couture@usherbrooke.ca; 4School of Rehabilitation, Faculty of Medicine, Université de Montréal, 2900 Bd Édouard-Montpetit, Montréal, QC H3T 1J4, Canada; nathalie.bier@umontreal.ca

**Keywords:** self-supervised learning, SimCLR framework, self-attention, sharpness-aware minimization (SAM), human activity recognition, ambient sensors, smart homes

## Abstract

Deep learning models have gained prominence in human activity recognition using ambient sensors, particularly for telemonitoring older adults’ daily activities in real-world scenarios. However, collecting large volumes of annotated sensor data presents a formidable challenge, given the time-consuming and costly nature of traditional manual annotation methods, especially for extensive projects. In response to this challenge, we propose a novel AttCLHAR model rooted in the self-supervised learning framework SimCLR and augmented with a self-attention mechanism. This model is designed for human activity recognition utilizing ambient sensor data, tailored explicitly for scenarios with limited or no annotations. AttCLHAR encompasses unsupervised pre-training and fine-tuning phases, sharing a common encoder module with two convolutional layers and a long short-term memory (LSTM) layer. The output is further connected to a self-attention layer, allowing the model to selectively focus on different input sequence segments. The incorporation of sharpness-aware minimization (SAM) aims to enhance model generalization by penalizing loss sharpness. The pre-training phase focuses on learning representative features from abundant unlabeled data, capturing both spatial and temporal dependencies in the sensor data. It facilitates the extraction of informative features for subsequent fine-tuning tasks. We extensively evaluated the AttCLHAR model using three CASAS smart home datasets (Aruba-1, Aruba-2, and Milan). We compared its performance against the SimCLR framework, SimCLR with SAM, and SimCLR with the self-attention layer. The experimental results demonstrate the superior performance of our approach, especially in semi-supervised and transfer learning scenarios. It outperforms existing models, marking a significant advancement in using self-supervised learning to extract valuable insights from unlabeled ambient sensor data in real-world environments.

## 1. Introduction

According to the United Nations World Social Report 2023, the global population aged 65 and older is projected to double over the next three decades, reaching 1.6 billion in 2050 [1]. Many countries have proposed “Aging in Place” to assist older adults to live in their own homes and communities safely, independently, and comfortably, regardless of age, income, or ability level [2,3,4]. Human activity recognition (HAR) based on sensors has made significant contributions to human-centric applications in smart homes for elderly care, encompassing home automation, ambient assisted living (AAL) systems, and remote monitoring [5,6]. Among the three main types of sensors used—cameras, wearable, and ambient sensors—each has its predominant advantages and limitations. While camera-based sensors raise privacy concerns within the home environment, and wearable sensors are considered intrusive solutions, the adoption of ambient sensors is growing [7]. Ambient sensors enable the collection of data streams to monitor older adults’ daily activities in real-world environments [8].

In recent decades, deep learning models, such as convolutional neural networks (CNNs) and long short-term memory (LSTM) networks, have made significant advances in HAR applications [9]. Wang et al. [10] mentioned that CNN models can extract features without prior knowledge, making them suitable for HAR tasks. LSTM networks, in particular, excel at capturing long-term dependencies and acquiring hierarchical representations from sequential ambient sensor data, a characteristic often observed in HAR [11,12].

However, achieving competitive performance with deep learning models in HAR requires a substantial amount of annotated data. Annotating activity data in real-world smart homes poses a significant challenge, as it is time-consuming and tedious for researchers and experts [13]. For instance, the Soutien à l’Autonomie des Personnes Âgées; Support for the Autonomy of Older Adults (SAPA) project, in collaboration with the Domotique et Informatique Mobile (DOMUS) laboratory, Centre de Recherche Institut Universitaire de Gériatrie de Montréal (CRIUGM), and Le Laboratoire d’Intelligence Ambiante pour la Reconnaissance d’Activités (LIARA) [14], has installed thirty-eight smart home frameworks in private residences in collaboration with three Canadian universities since February 2019. The project aims to support cognitive assistance and telemonitoring of frail elderly individuals with cognitive deficits using Internet of Things (IoT) technologies deployed in senior living communities [15]. Over four years, the project has generated massive amounts of ambient sensor data. However, the lack of ground truth data hinders the effectiveness of deep learning models.

Recently, self-supervised learning (SSL) has gained popularity in various fields such as computer vision (CV), natural language processing (NLP), speech, and audio processing [16]. SSL becomes attractive because it eliminates the need for large-scale annotated datasets, which can be expensive and time-consuming to create [17]. In the context of time-series classification tasks like HAR, contrastive learning has emerged as a prominent technique in SSL. The SimCLR framework [18] has proven to be a powerful transfer-based encoder for learning feature representations from unlabeled sensor data in HAR [19]. Several researchers have proposed novel models, including SimCLRHAR [20], CSSHAR [19], and ClusterCLHAR [21], which are based on the SimCLR framework. These models have demonstrated impressive accuracy in HAR tasks using sensor data from smartphones and smartwatches. However, SimCLR framework-based models in HAR using ambient sensor data from real-world smart homes are still relatively uncommon.

Motivated by the success of SSL in HAR tasks with limited labeled data, and the effectiveness of LSTM models and self-attention in handling time-series sequence data [22,23], we propose a model called AttCLHAR that integrates self-attention, SimCLR, and sharpness-aware minimization (SAM). This model is specifically tailored for ambient sensor data in real-world smart homes. This journal paper is an extended version of our previous work [24], with the main extensions being: (1) an enhancement of the encoder to incorporate self-attention and LSTM for identifying the most relevant parts of the input sequence; and (2) the use of SAM in both the linear and fine-tuning scenarios to improve generalization, inspired by [25]. In summary, the principal contributions of this work are as follows:A novel adapted contrastive AttCLHAR framework is proposed, which incorporates two layers of one-dimensional CNN, one layer of LSTM, and a self-attention layer as the encoder. This encoder is designed to learn robust representations from time-series data. The proposed framework is evaluated on real-world smart home datasets, demonstrating its effectiveness in capturing and recognizing human activities in this context.Extensive experiments were conducted on real-world smart home datasets to evaluate the proposed model’s performance in HAR and compare it with three other models: SimCLR [24], SimCLR with SAM, and SimCLR with the self-attention layer (AttSimCLR). The results demonstrate that all the models’ representations are highly effective for downstream tasks, and AttCLHAR performs better, especially in semi-supervised learning.Transfer learning experiment shows that the robust features extracted from the Aruba-2 dataset [26] can be successfully applied to the Aruba-1 dataset [26], achieving competitive performance when compared to the semi-supervised learning approach specifically applied to the Aruba-1 dataset.

The paper is structured as follows: Section 2 presents the related work on HAR on ambient sensors, SSL, SimCLR for HAR, and self-attention mechanism. The methodology of the proposed approach is introduced in Section 3. Comprehensive experiments on CASAS datasets are detailed in Section 4, and the results and discussion are presented in Section 5. Finally, Section 6 provides the conclusion.

## 2. Related Work

This section introduces the related work on HAR from ambient sensors and SSL, explicitly focusing on the SimCLR framework for HAR and the incorporation of the self-attention mechanism.

### 2.1. Human Activity Recognition from Ambient Sensors

Most older adults wish to age in place, even with incapacities. To support this, unobtrusive ambient sensors have emerged as the preferred solution for monitoring the daily activities of older adults living alone [27]. The rise of remote elderly home monitoring has spurred numerous studies on HAR as a fundamental aspect of AAL [28].

Gochoo et al. [27] proposed a deep convolutional neural network (DCNN) for an unobtrusive activity recognition classifier using anonymous binary sensors, such as passive infrared motion and door sensors. Although the DCNN model effectively extracted features from binary activity images, it overlooked temporary dependencies between activities, such as “Wash dishes” mostly occurring after “Meal preparation”. Hybrid models, comprising a combination of LSTM and CNN [29], were employed to address this limitation. These models considered the importance of upcoming short-term sensor data and the continuity from past activities in preceding long-term sensor data. The combination model successfully learned the spatiotemporal dependencies of the fused features. However, instead of highlighting crucial features from the sensor data, the model treated all the hidden states as equally important. Fahad et al. [30] performed activity recognition by applying a probabilistic neural network (PNN) to pre-segmented activity data obtained from sensors deployed at various locations in CASAS smart home datasets, including Aruba and Milan [26]. However, the achieved performance was not competitive.

Due to the lack of annotations in real-world smart homes, Gupta et al. [31] proposed using kernel density estimation (KDE) for preprocessing data. This is combined with t-distributed stochastic neighbor embedding and uniform manifold approximation and projection for visualizing changes, thereby discovering patterns in unlabeled sensor data to track user activity changes. While this model is capable of detecting patterns in activities over longer time frames, precise HAR on unlabeled sensor data remains essential for AAL.

### 2.2. Self-Supervised Learning

Self-supervised learning (SSL) is a powerful technique that allows the extraction of feature representations by training encoders on massive amounts of unlabeled data. The training process comprises two main stages: a pretext task and downstream tasks, where the knowledge learned from the pretext models is transferred to specific tasks by fine-tuning the features [17].

Self-supervised learning can be categorized into two main approaches: contrastive and non-contrastive. Contrastive learning aims to distinguish between similar and dissimilar data within the input data [32,33]. For instance, contrastive predictive coding (CPC) [34] learned representations by predicting future samples in the latent space, using powerful autoregressive models for generating predictions and employing a probabilistic contrastive loss. Momentum contrast (MoCo) [35] utilized a momentum encoder to extract representations from negative samples using a memory bank. SimCLR [18] proposed a large batch size of negative samples without a memory bank, improving contrastive learning performance. In contrast, non-contrastive learning models learn from positive examples only, consisting of original and augmented data. Examples of non-contrastive approaches include bootstrap your own latent (BYOL) [36], simple Siamese (SimSiam) [37], Barlow twins [38], and variance-invariance-covariance regularization (VICReg) [39], which present promising results from unlabeled data without relying on contrastive loss or pairwise comparisons.

Given the recent popularity of SSL in HAR research [19,40], this paper emphasizes SSL approaches for HAR, as they have demonstrated their effectiveness in capturing meaningful representations from unlabeled sensor data for HAR.

### 2.3. SimCLR for Human Activity Recognition

SimCLR [18] is a highly advanced self-supervised contrastive learning model, widely used in HAR research. It offers a straightforward framework for learning visual representations through contrastive learning. This is achieved by leveraging data augmentation, a nonlinear projection head, and a large batch size, all without the need for a memory bank.

An adapted version of SimCLR [40] was applied to HAR, specifically for recognizing activities such as walking down and up stairs, jogging, sitting, and standing. The results demonstrated competitive performance, especially in healthcare applications, even with limited labeled data. Another study introduced CSSHAR [19], which replaced SimCLR’s backbone network with a custom transformer encoder to enhance feature representations extracted from unlabeled sensory data in HAR. ClusterCLHAR [21], following the SimCLR framework, proposed a novel contrastive learning approach for HAR by incorporating negative selection through clustering. The experimental results of ClusterCLHAR show competitive performance in both self-supervised and semi-supervised learning for HAR.

Although SimCLR has been extensively used for HAR based on smartphones or wearable sensors, there is currently a lack of research focusing on applying self-supervised learning to HAR based on ambient sensor data.

### 2.4. Self-Attention Mechanism

The attention mechanism has been widely employed in deep learning models. It was originally introduced by [41] and formed the basis for subsequent developments. Zeng et al. [42] proposed temporal and sensor attention for HAR, focusing on important signals and sensor modalities. This approach helps to interpret recurrent networks and gain insights into model behavior. Raffel and Ellis [43] asserted that attention mechanisms could address certain long-term memory problems. The work by Vaswani et al. [44] demonstrated that Transformers rely solely on attention mechanisms, eliminating the need for recurrence and convolutions.

Mahmud et al. [45] introduced a self-attention-based neural network model that eschews recurrent architectures, employing different attention mechanisms to generate higher-dimensional feature representations for classification. Researchers [23,46] applied attention to recurrent convolutional attention networks or LSTM networks for sensor-based human activity recognition. The self-attention mechanism proves effective in capturing important features in ambient sensor sequences.

Wang et al. [47] proposed a novel deep multi-feature extraction framework based on the attention mechanism (DMEFAM). Singh et al. [22] utilized a self-attention mechanism to learn crucial time points for decoding human activity. This paper leverages self-attention in the AttCLHAR model to extract important information from sequential time series ambient sensor data.

## 3. Methodology

This section elaborates on the components of the proposed AttCLHAR framework, which extends the SimCLR approach [18,19,40] for HAR. The framework comprises two primary stages: pre-training and fine-tuning. Figure 1 illustrates the overall architecture of the model.

### 3.1. Pre-Training

The pre-training stage aims to extract representative features from extensive amounts of unlabeled data and typically involves the following key components.

Preprocessing of unlabeled ambient sensor data involves extracting multiple relevant features or performing necessary transformations on raw, unlabeled sensor data. It aims to ensure the data are in an appropriate format for subsequent processing and analysis.

The augmentation module is a critical component of the proposed framework, as it transforms the preprocessed ambient sensor data into two distinct views, facilitating data augmentation. The views from the same sample are considered positive pairs, whereas views from different samples within the same batch are considered negative pairs. As emphasized in [18], proper data augmentation is essential for defining effective predictive tasks in self-supervised learning. Given a sample *x*, the argumentation model T produces two views, $x_{i}$ and $x_{j}$. The details of augmentation selection will be discussed in Section 4.2.2.

A neural network-based encoder f(·) generates embedded representations $s_{i}$ and $s_{j}$ in the latent space from augmented input data $x_{i}$ and $x_{j}$. Due to the suitability of 1D convolution for HAR using sensor data [48], the encoder comprises two layers of one-dimensional CNN with ReLU activation, incorporating dropout after each convolutional layer, one LSTM layer, and an attention mechanism layer. The details of each of these layers and the training are presented in Section 4.2.3. The purpose of the attention mechanism is to focus on essential features rather than the entirety of the LSTM output. L2 regularization has been applied to mitigate overfitting. After the LSTM model generates the hidden state representation $\langle h_{t}\rangle$, an attention layer uses an attention matrix *A* to capture the similarity of any sequence while considering all neighboring sequences. Here, $\alpha_{t,t^{\prime }}\in A$ represents the attention weight between hidden state representations $h_{t}$ and $h_{t^{\prime }}$ at timesteps *t* and $t^{\prime }$, respectively. The utilized attention mechanism is multiplicative attention as follows [49,50,51]:(1)$$e_{t,t^{\prime }}=h_{t}^{T}W_{a}h_{t^{\prime }}+b_{a}$$
(2)$$\alpha_{t,t^{\prime }}=softmax\left(e_{t,t^{\prime }}\right)$$
(3)$$c_{t}=\sum_{t^{\prime }=1}^{T}\alpha_{t,t^{\prime }}h_{t^{\prime }}$$
where $e_{t,t^{\prime }}$ represents attention scores, $W_{a}$ denotes the weight matrices for the non-linear combination of hidden states, and $b_{a}$ is the bias. Subsequently, the attention-focused context vector $c_{t}$ at timestep *t* is computed as the sum of the hidden states $h_{t^{\prime }}$ at timesteps $t^{\prime }$ and their respective attention weights $\alpha_{t,t^{\prime }}$ to the current hidden state $h_{t}$, as outlined in [49]. At the end of the encoder for temporal sensor data, a 1D global max-pooling operation is then performed to extract the maximum vector over the steps dimension.

The projection head is composed of three fully connected neural network layers. It receives latent space representations as input and projects them into a different space. This projection facilitates the calculation of the contrastive loss, which is essential for training the self-supervised learning model. The expression g(·) represents the neural network projection head to map the latent representation to another space for contrastive loss. A multilayer perceptron (MLP) with two hidden layers was used to obtain $z=g\left(s\right)=W^{\left[2\right]}\sigma \left(W^{\left[1\right]}\sigma \left(W^{\left[0\right]}s\right)\right)$, where σ is a ReLU activation function and *W* is weight of the MLP. The projected representations generated by the projection head are subsequently utilized to calculate the NT-Xent loss.

A contrastive loss function is employed to maximize the similarity between positive data pairs and minimize the similarity scores for negative data pairs [19]. The NT-Xent loss (the normalized temperature-scaled cross-entropy loss) has been used in previous work [18,20], and is defined as follows:(4)$$l\left(i,j\right)=-\log\frac{\exp(sim(i,j)/\tau )}{\sum_{k=1}^{2N}I_{\left[k\neq i\right]}\exp\left(sim\left(i,k\right)/\tau \right)}$$
(5)$$L=\frac{1}{2N}\sum_{k=1}^{N}\left[l\left(2k-1,2k\right)+l\left(2k,2k-1\right)\right]$$
In the given equation, τ is a hyperparameter called a temperature parameter to scale the similarity scores between positive and negative pairs; sim(i,j) is a cosine similarity function; *N* is a batch size; and $I_{\left[k\neq i\right]}$ is an indicator function, and the value is 1 when *k* is not equal to *i*. All the positive pairs l(2k−1,2k) and l(2k,2k−1) in the mini-batch calculate the loss. The 2k−1 and 2k are positive pairs, and 2k−1 combined with other views are negative pairs. Then, the model is passed SAM to improve model generalization by penalizing loss sharpness. When a given loss function $L:W\times X\times Y\to R_{+}$ and training loss $L_{S}$, SAM will get parameter *w* in the neighborhood ρ that has low loss as well as low curvature by optimizing the min-max objective, which is given by [25,52]:(6)$$L_{S}^{SAM}\left(w\right)\overset{\Delta }{=}\max_{{∥\epsilon ∥}_{p}\leq \rho }L_{S}\left(w+\epsilon \right)$$
(7)$$\min_{w}L_{S}^{SAM}\left(w\right)+\lambda {∥w∥}_{2}^{2}$$
where ρ≥0 is a hyperparameter and $\rho \in \left[1,\infty \right]$; $\lambda {∥w∥}_{2}^{2}$ is a standard L2 regularization term with a hyperparameter ρ. In Equation (Equation 6), the *p*-norm is generalized slightly from an L2-norm in the maximization over ϵ [52].

### 3.2. Fine-Tuning

Fine-tuning is conducted on preprocessed, sparsely labeled ambient sensor data using the pre-trained encoder for feature extraction. During this phase, the previously frozen encoder weights are unfrozen to allow updates. The projection model used in pre-training is then discarded. For the downstream task of activity classification, a multilayer perceptron (MLP) prediction model with two layers is used. This model employs a ReLU activation function for the hidden layer and a Softmax activation function for multiclass activity classification, along with a cross-entropy loss function.

## 4. Experimental Setup

This section provides an overview of the experimental datasets and details the implementation of the experiments conducted to evaluate the proposed approach.

### 4.1. Datasets

The experiment utilized three smart home datasets: Aruba-1, Aruba-2, and Milan, publicly available from the CASAS project [26]. These datasets involve motion sensors, door closure sensors, and temperature sensors, with the floor layout depicted in Figure 2. Temperature sensors were excluded from feature extraction and model training. Both motion sensors and door closure sensors used the same feature extraction method in the model.

The Aruba-1 dataset is an annotated collection of sensor data gathered over approximately seven months in a resident’s smart home, shown in Figure 2a. The resident conducted daily activities without supervision or scripted scenarios, with occasional visits from her children. This dataset comprises 6477 instances annotated with 11 activity categories, including *Meal_Preparation*, *Relax*, *Eating*, *Work*, *Sleeping*, *Wash_Dishes*, *Bed_to_Toilet*, *Enter_Home*, *Leave_Home*, *Housekeeping*, and *Resperate*. The raw sensor data includes information such as *date*, *time*, *sensor ID*, *sensor state*, *activity*, and *activity status* shown in Table 1. To facilitate evaluation and due to the limited number of instances for the *Resperate* activity, this activity, along with sensor data lacking activity labels in the Aruba-1 dataset, was excluded from the experiment.

In contrast, the Aruba-2 dataset is an unannotated collection of raw ambient sensor data spanning nearly one year. This dataset maintains the same testbed layout and ambient sensor positions as the Aruba-1 dataset, as illustrated in Figure 2a.

The Milan dataset comprises labeled sensor data collected over three months from a single person living with a pet, as depicted in Figure 2b. During this period, the resident’s children and grandchildren regularly visited. As outlined in [11], the dataset initially included fifteen distinct activity categories, which were later consolidated into ten activities of daily living (ADLs) for consistent result comparison. The selected ADL categories, commonly employed to assess an individual’s functional health within a clinical context, are presented in Table 2. For instance, *Dining Rm Activity* corresponds to *Eating*, and *Kitchen Activity* maps to *Meal_Preparation*. The ten ADLs categories used in the experiment include: *Meal_Preparation*, *Relax*, *Eating*, *Work*, *Sleeping*, *Bed_to_Toilet*, *Leave_Home*, *Medication*, *Bathing*, and *Other*. The *Other* category is assigned to ambient sensor data when the resident is not engaged in any specific activity.

### 4.2. Implementation Details

This section provides the implementation details for each component of the proposed model AttCLHAR shown in Figure 1.

#### 4.2.1. Data Preprocessing

The data preprocessing steps were consistent across the three datasets (Aruba-1, Aruba-2, and Milan). To ensure the interpretability of the raw sensor events presented in Table 1, several cleaning procedures were performed, including handling missing values, correcting format errors, and removing duplicate data. The locations of motion sensors and contact sensors were mapped in Table 3 for Aruba-1, Aruba-2 and Milan datasets.

The feature extraction process was based on the AL Activity Learning–Smart Home approach [53], which involved generating a feature vector for each non-overlapping sensor sequence with a fixed window length of 30. The number of features, relative to the total sensors in the dataset, is presented in Table 4. The *num_sensors* represents the total number of sensors in the dataset. In particular, the Aruba-1 and Aruba-2 datasets, which collectively had 35 sensors, produced 84 features. Conversely, the Milan dataset, equipped with 33 sensors, resulted in 80 features.

The preprocessed data were then divided into fixed windows. Taking the Aruba-1 dataset as an example, the preprocessed sensor data, with a shape of (731106, 84) encompassing 84 features, was partitioned into windowed datasets using a sliding window (10, 84) with a 50% overlap. This process resulted in three-dimensional data with dimensions (samples, sliding window size, features) as (146220, 10, 84). The same data processing procedure was applied to the Aruba-2 and Milan datasets. Subsequently, the dataset was split into training, evaluation, and test sets based on the experiment’s requirements. The proportions of these splits may vary depending on the specific experiment being conducted. Finally, the sensor features were standardized to have a zero mean and a standard deviation of 1. The normalization process ensures that the features are on a consistent scale, preventing the influence of different feature scales on the models’ performance.

#### 4.2.2. Augmentation Selection

The augmentation module applied five selected augmentation techniques mentioned in [40,54] to the ambient sensor data. These techniques included random signal inverting, reversing, time warping, random noise, and scaling. Experiments were conducted on the Aruba-1 dataset to identify the most effective augmentation function using the SimCLR framework [24] and AttCLCHAR. The performance of the augmented dataset was evaluated in both linear evaluation and fine-tuning scenarios, using F1 score and accuracy as evaluation metrics. The results are presented in Table 5. The performance analysis indicates that the scaling augmentation method with a scale of 0.1, in both linear and fine-tuning scenarios, outperforms the other augmentation methods on both models, SimCLR and AttCLCHAR.

#### 4.2.3. Other Components

Table 6 shows the hyperparameters for the proposed model, AttCLCHAR. The input data for the three datasets differed, with dimensions (146220, 10, 84), (400083, 10, 84), and (83325, 10, 80) for Aruba-1, Aruba-2, and Milan, respectively. The one-dimensional CNN utilized two layers with a fixed kernel size of 3. The number of output channels for these layers was set to 32 and 64, respectively. L2 regularization with a learning rate of 1 $\times \;10^{-4}$ was applied. Additionally, a dropout rate of 0.1 was added after each convolutional layer. Following the convolutional layers, the model incorporated a one-layer LSTM with 64 units. The attention layer employed multiplicative attention and a regularizer with a strength of 1 $\times \;10^{-4}$.

The projection head consisted of three fully connected layers with dimensions of 256, 128, and 64, respectively. The outputs from the first two layers are processed through the ReLU activation function. The projected representations generated by the projection head are subsequently used to calculate the NT-Xent loss. For the NT-Xent loss calculation, a temperature parameter of 0.1 was employed. The model parameters were optimized using stochastic gradient descent (SGD) [55] with cosine decay, as described in [40]. This optimization technique gradually reduced the learning rate during training, allowing the model to converge effectively. The initial learning rate was set to 0.1, and the decay steps were set to 1000. The pre-training batch size was 512, and the model was trained for 100 epochs. In the linear evaluation setup, the model parameters were optimized using the Adam optimizer with an initial learning rate of 0.001, as mentioned in [20]. A batch size of 128 was used during the linear evaluation, which was performed for 300 epochs. For the fine-tuning phase, the Adam optimizer was employed with an initial learning rate set to 0.001. A batch size of 128 was utilized for training. Both linear evaluation and fine-tuning used cross-entropy as the loss function, and the evaluation metrics used were F1 score and accuracy.

## 5. Results and Discussion

This section presents experimental results of contrastive learning, semi-supervised learning, and transfer learning. The experiment was executed on an Intel(R) Xeon(R) W-2133 CPU @ 3.60 GHz, utilizing the TensorFlow library. The pre-training process involved a model size of 109,217 total trainable parameters (426.63 KB) for Aruba-1 and Aruba-2, and 108,833 parameters (425.13 KB) for Milan. Taking the Aruba 1 training as an example, the pre-training, which consisted of 100 epochs, took approximately two hours. Linear evaluation with 300 epochs took around ten minutes. Fine-tuning with 200 epochs required about fifteen minutes.

### 5.1. Contrastive Learning Scenario

This section evaluates the quality of contrastive learning features in the latent space with a linear evaluation protocol on the Aruba-1 and Milan datasets. The encoder was first pre-trained on all the Aruba-1 and Milan datasets without activity labels. After obtaining the representation vectors from the encoder, the encoder weights were frozen, the projection head was dropped, and a one-layer dense classifier was added. This classifier was trainable and used for activity classification.

Different proportions of training data were evaluated in the linear evaluation scenario using the Aruba-1 dataset. The training set was divided into proportions of 1%, 5%, 10%, 20%, and 30%, while the remaining data was used for evaluation and testing. The performances of four models, SimCLR [24], SimCLR with SAM, AttSimCLR (SimCLR with attention), and AttCLCHAR, were compared using F1 score and accuracy. The results are presented in Table 7. The Aruba-1 linear evaluation indicates that even with just 1% of the training data, all models achieved F1 scores and accuracy above 80.67% and 80.87%, respectively. When the training data proportion increased to 20% or more, all models surpassed 85% in both F1 score and accuracy except SimCLR with SAM. Although SimCLR performed better with 1% and 5% proportions, AttCLCHAR outperformed as the label proportions increased. SimCLR and AttSimCLR showed similar performance for all the different training proportions. These findings suggest that the proposed model, AttCLCHAR, can learn meaningful feature representations even without the availability of activity labels and has better performance compared to other models.

For the evaluation of the Milan dataset, the same model settings were applied as for the Aruba-1 dataset, except for the training set proportions. The training set proportions of the Milan dataset were set to 10%, 20%, 30%, 40%, and 50%. This adjustment was made because linear evaluation performance tended to be lower when the training set constituted a small proportion of the data. The results are shown in Table 8, where the F1 score and accuracy of Milan’s linear evaluation for different ratios range between 59.74% and 64.39% for SimCLR and 61.28% and 64.17% for AttCLCHAR. The model AttCLCHAR performs better when the proportions are below 40%, but the model SimCLR is better when the proportion reaches 50%. However, SimCLR with SAM outperforms other models in all the label fractions. These performances of the linear evaluation on the Milan dataset are not as good as the Aruba-1 dataset. This discrepancy can be attributed to several factors mentioned in [12] for the Milan dataset, such as the presence of a resident living with a pet, which introduces more noise, duplicate data in the dataset, and improper ordering of sensor activations during certain periods. These factors can affect the activity recognition model’s performance and result in lower F1 scores and accuracy.

Figure 3 visualizes the feature representations for Aruba-1 and Milan using the t-SNE algorithm [56] to project high-dimensional representations into two-dimensional space. These representations were generated by the outputs of the AttCLCHAR model’s pre-trained encoder using unlabeled data and were then evaluated using 20% of the total data. Aruba-1 shows superior visualization representation compared to Milan, aligning with the F1 scores and accuracy results presented in Table 7 and Table 8. Although the model effectively learns the majority of activities, imbalanced activity impacts its performance, a challenge that will be tackled in future work.

Overall, the proposed model demonstrates its ability to learn useful representations through extensive evaluation based on the above experiments. The F1 scores exhibit minimal variations for different proportions of training data in the linear evaluation of the Aruba-1 and Milan datasets, with only approximately a 2% or less difference. This suggests that contrastive learning features can effectively represent the underlying features of the data.

### 5.2. Semi-Supervised Learning Scenario

This section focuses on the fine-tuning of the proposed encoder and prediction MLP model using a small amount of labeled data from the Aruba-1 and Milan datasets.

For the Aruba-1 dataset, we compared the performances of four models: SimCLR [24], SimCLR with SAM, AttSimCLR, and AttCLCHAR, shown in Table 9. Although SimCLR exhibits superior performance with an F1 score of 89.16% with just 1% labeled data, AttCLCHAR outperforms it when the label proportion reaches 5% and beyond, surpassing the other three models. Notably, all four models achieve an F1 score of 88.20% and above, even with only 1% labeled data. Remarkably, AttCLCHAR demonstrates an impressive 6% performance increase with just 1% labeled data compared to the linear evaluation results presented in Table 7. These findings underscore the efficacy of AttCLCHAR, yielding competitive results even with a limited amount of labeled sensor data.

Subsequently, AttCLCHAR was compared with other models, including the probabilistic neural network (PNN) for supervised learning [57], known for its proficiency with sparse and noisy sensor data. Additionally, comparisons were made with a deep neural network (DNN) [58] and a bidirectional long short-term memory (Bi-LSTM) neural network tailored for recognizing daily activities [59]. Further, we assessed a Transformer-based semi-supervised learning (SSL) approach that integrated multiple sensor sequence inputs for HAR [57]. The results, detailed in Table 10, demonstrate the competitive performance of AttCLCHAR. Even with only 10% labeled activities, AttCLCHAR achieves an impressive F1 score of 95.60%, surpassing the supervised learning models that necessitate the entire set of labels. Upon increasing the label rate to 20%, AttCLCHAR exhibits a remarkable F1 score of 97.04% and an accuracy of 97.02%, outperforming not only the supervised learning models but also the Transformer-based SSL approach.

For the Milan dataset, we maintained consistency by applying the same fine-tuning settings as in the Aruba-1 experiment. Additionally, a similar approach was utilized as described in [11,12] for grouping activities during data preprocessing. Subsequently, the performances of SimCLR, SimCLR with SAM, AttSimCLR, and AttCLCHAR models were compared, as shown in Table 11. The results demonstrate that all the models can achieve an F1 score and accuracy exceeding 88.97% in the fine-tuning phase when the training set comprises approximately 20% or more of the data. AttSimCLR exhibits superior performance with label proportions of 20% and 30%, whereas the model AttCLCHAR outperforms the other three models when the label proportion reaches 40%.

The model AttCLCHAR was then compared with four supervised learning models, including PNN [30], a HAR algorithm based on wide time-domain convolutional neural network and multienvironment sensor data (HAR_WCNN) [60], LSTM [11], and Ens2-LSTM, which combined the output of a Bi-LSTM and unidirectional LSTM (Uni-LSTM). The detailed results are shown in Table 12. The experimental results highlight the competitive performance of AttCLCHAR compared to other supervised learning models, as evidenced by an F1 score of 95.21% and an accuracy of 95.19% using 40% labeled activity.

Figure 4 compares the performance improvement from the linear scenario to the fine-tuning scenario, presenting F1 score and accuracy for Aruba-1 and Milan using the AttCLCHAR model. When the label fraction is higher, the performance improvement from linear evaluation to fine-tuning is more significant compared to scenarios with a lower label fraction. In comparison to the improved performance from fine-tuning to linear evaluation for Aruba-1 and Milan, Milan exhibits a more significant enhancement, with an increase exceeding 31.8% when the training proportion reaches 30%. This greater improvement could be attributed to the challenges posed by the lower quality of the Milan dataset, which may affect the effectiveness of contrastive learning. However, the fine-tuning scenarios have demonstrated their effectiveness in refining and optimizing the parameters of the pre-trained encoder model, resulting in notable performance improvements for both the Aruba-1 and Milan datasets.

Furthermore, it demonstrates that as the proportion of the training set for Aruba-1 increases from 1% to 10%, the F1 score of fine-tuning improves by a higher percentage. Similarly, Milan shows more performance improvement from 10% to 30% than from 30% to 50% for fine-tuning. This finding suggests that high accuracy can be achieved even with a small number of labeled samples.

These evaluations provide strong evidence that the proposed model achieves competitive performance by leveraging an encoder with limited annotated data. This finding is particularly significant, as it addresses the challenge of limited annotation availability in real-world environments. The results highlight the potential of the model to effectively learn from unlabeled data and overcome the limitations of relying solely on annotated datasets.

### 5.3. Transfer Learning Scenario

Transfer learning involves transferring knowledge from one dataset to another. The transfer learning performance was evaluated on the Aruba-1 and Aruba-2 datasets using linear evaluation and fine-tuning settings.

The proposed approach followed two steps for transfer learning. Firstly, the encoder was pre-trained on the Aruba-2 dataset using self-supervised learning. The encoder weights were then frozen, and the projection head was removed. Then, the downstream tasks were evaluated with linear evaluation and fine-tuning on Aruba-1. For the linear evaluation, only the output MLP model was trainable. It was trained on top of the pre-trained encoder’s features. For fine-tuning, the entire network, including the pre-trained encoder, was trainable.

The experiments involved varying proportions of the training set, aligning with the linear evaluation scenario using SimCLR and AttCLCHAR models for the Aruba-1 dataset, as detailed in Table 7. Evaluation metrics, including F1 score and accuracy, are presented in Table 13.

In the transfer learning scenario, the linear evaluation results for the SimCLR model showcase an enhanced F1 score and accuracy compared to all linear evaluations conducted on the Aruba-1 dataset, as shown in Table 7. This improvement can be attributed to the greater availability of unlabeled ambient sensor data in the Aruba-2 dataset compared to Aruba-1, which is used for pre-training. The enhanced representation learned through pre-training transfers better to downstream tasks. While AttCLCHAR’s linear evaluation performances may not match SimCLR, its fine-tuning performances surpass SimCLR when training proportions exceed 5%, demonstrating competitive performance against AttCLCHAR’s fine-tuning results in Table 9. These results underscore the promising performance of the proposed AttCLCHAR model. It indicates that pre-trained features on one unlabeled dataset can be effectively optimized for downstream tasks in another dataset with limited labeled data. This observation highlights the potential of training models in real-world settings, leveraging abundant unlabeled data and annotations from other datasets to enhance overall performance.

## 6. Conclusions

A novel approach, AttCLCHAR, is proposed based on the SimCLR framework with self-attention and SAM for HAR using ambient sensor data from smart homes. The model combines CNN, LSTM, and self-attention networks in the encoder to capture local and global features in time-series data. Extensive experiments on three publicly available smart home datasets compare the performances of four models: SimCLR, SimCLR with SAM, AttSimCLR, and AttCLCHAR. The results demonstrate the robust performance of AttCLCHAR in semi-supervised learning and transfer learning, outperforming other models and state-of-the-art deep learning models. This effectiveness allows the model to learn meaningful contrastive representations from a wide range of real-world unlabeled data.

Future research can enhance the proposed approach by addressing specific challenges in real-world applications. One such challenge is dealing with imbalanced daily activities in real-world datasets, where certain activities may occur more frequently than others. It is crucial to develop strategies to address these imbalances, ensuring that the model performs effectively across all activities, including those less represented. Additionally, the model can be tested on smart homes with different layouts and sensor installations to evaluate its transfer learning capabilities across diverse environments. As part of future work, we plan to implement and evaluate the model on unlabeled ambient data from the SAPA project of the DOMUS laboratory [14]. The large volume of unlabeled data is essential to ensure a better representation learning during pre-training. This will enhance the model’s scalability and feasibility for real-world applications.

By addressing the identified areas for improvement, such as considering imbalanced daily activities and testing the model in smart homes with various layouts and sensor installations, the AttCLCHAR can be further enhanced for human activity recognition in different smart home environments. This research has the potential to advance the field and contribute to the development of more effective and adaptable ambient sensing systems, ultimately benefiting the aging population and improving their quality of life in smart home environments.

## Figures and Tables

**Figure 1 sensors-24-00884-f001:**
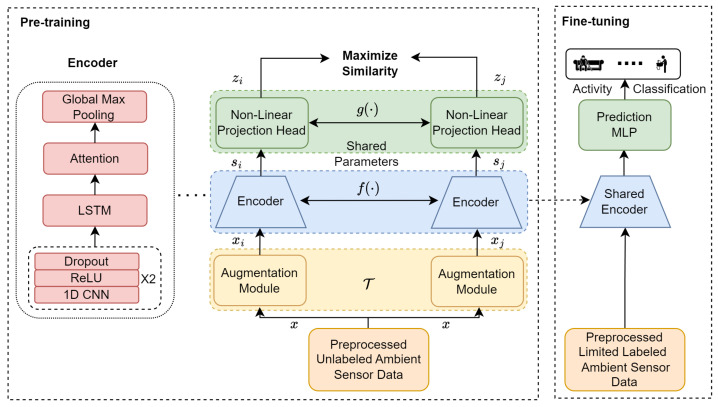
The architecture of the AttCLHAR framework for HAR.

**Figure 2 sensors-24-00884-f002:**
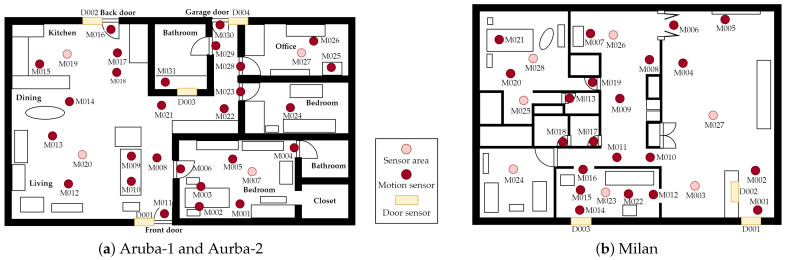
Layout of door and motion sensors in Aruba and Milan testbeds.

**Figure 3 sensors-24-00884-f003:**
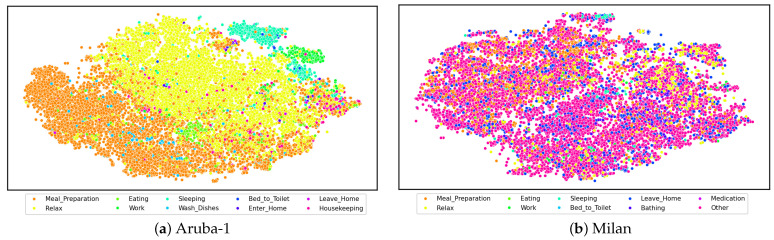
Comparative visualization of Aruba-1 and Milan feature representations using t-SNE.

**Figure 4 sensors-24-00884-f004:**
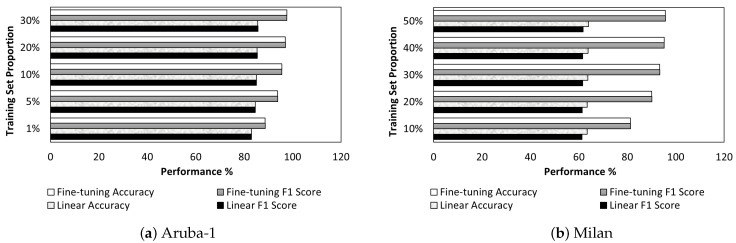
The comparison of performance improvement from the linear scenario to the fine-tuning scenario with F1 score and accuracy for Aruba-1 and Milan using the AttCLCHAR model.

**Table 1 sensors-24-00884-t001:** Aruba-1 raw ambient sensor data.

Date	Time	Sensor ID	Sensor State	Activity	Activity Status
2010-11-04	08:33:52.929406	M018	ON	Meal_Preparation	begin
2010-11-04	08:33:53.911115	M019	OFF		
2010-11-04	08:33:54.892943	M017	ON		
2010-11-04	08:33:57.253739	M018	OFF		
2010-11-04	08:35:43.498813	M019	OFF		
2010-11-04	08:35:44.246428	M019	ON		
2010-11-04	08:35:45.822482	M018	OFF	Meal_Preparation	end

**Table 2 sensors-24-00884-t002:** The activity categories within the Milan dataset were reorganized to create new categories of activities [24].

**Original Activity**	Read Watch TV	Morning Meds * Eve Meds *	Master Bathrm Guest Bathrm	Desk Activity Chores	Meditate Master Bedroom	Dining Rm * Activity	Kitchen Activity
**New Activity**	Relax	Medication	Bathing	Work	Other	Eating	Meal Preparation

* Rm or rm: room; Meds: medications.

**Table 3 sensors-24-00884-t003:** Locations of motion and contact sensors in Aruba and Milan.

Aruba-1 and Aruba-2		Milan	
Location	Core Sensors	Location	Core Sensors
Kitchen	M015, M016, M017,	Kitchen	M012, M014, M015, M016,
	M018, M019		M022, M023, D003
Living room	M009, M010, M012, M013, M020	Living room	M006, M008, M026,
Bedroom 1	M001, M002, M003, M005, M006, M007	Bedroom 1	M019, M020, M021, M028
Bedroom 2	M023, M024	Bedroom 2	M024
Bathroom 1	D003, M029, M031	Bathroom 1	M013, M025
Bathroom 2	M004	Bathroom 2	M017, M018
Dining	M014	Dining	M003, M027
Office	M025, M026, M027, M028	Office	M007
Hall 1	M008	Hall 1	M010, M011
Hall 2	M021, M022	Hall 2	M009
Front door	M011, D001	Front door	M001, M002, D001, D002
Back door	D002	Reading room	M004, M005
Garage door	M030, D004	-	-

**Table 4 sensors-24-00884-t004:** Extracted features from raw ambient sensor data [53].

Feature Number	Feature
1	The time of the last sensor event in a window (hour)
2	The time of the last sensor event in a window (seconds)
3	The day of the week for the last sensor event in a window
4	The window size in time duration
5	The time since the last sensor event
6	The dominant sensor in the previous window
7	The dominant sensor is two windows back
8	The last sensor event in a window
9	The last sensor location in a window
10	The last motion sensor location in a window
11	The window complexity (entropy calculated from sensor counts)
12	The change in activity level between two halves of a window
13	The number of transitions between areas in a window
14	The number of distinct sensors in a window
15 ∽ num_sensors+14	The counts for each sensor
num_sensors+15 ∽ 2×num_sensors+14	The time since the sensor last fired

**Table 5 sensors-24-00884-t005:** Comparison of experimental results of augmentation methods on the Aruba-1 dataset with linear evaluation and fine-tuning using SimCLR and AttCLCHAR.

Augmentation Method	SimCLR [24] Linear Evaluation		SimCLR [24] Fine-Tuning		AttCLCHAR Linear Evaluation		AttCLCHAR Fine-Tuning	
	F1 Score (%)	Accuracy (%)	F1 Score (%)	Accuracy (%)	F1 Score (%)	Accuracy (%)	F1 Score (%)	Accuracy (%)
Inverting	80.45	80.86	95.25	95.24	81.57	81.77	96.86	96.85
Reversing	80.95	81.25	95.24	95.23	81.53	81.67	96.83	96.82
Time Warping	72.92	74.21	95.08	95.07	81.03	81.14	97.00	96.99
Random Noise	78.65	79.00	95.39	95.38	79.41	79.77	96.83	96.81
Scaling	81.42	81.54	95.56	95.54	85.41	85.36	97.04	97.02

**Table 6 sensors-24-00884-t006:** Parameter setting of proposed model AttCLCHAR for Aruba-1, Aruba-2, and Milan.

Parameters	Aruba-1	Aruba-2	Milan
Features	84	84	80
Input data	(146,220, 10, 84)	(400,083, 10, 84)	(83,325, 10, 80)
Layer 1 Conv 1D	Filters 32; kerne size 3; ReLU; L2 regularizer; dropout 0.1		
Layer 2 Conv 1D	Filters 64; kernel size 3; ReLU; L2 regularizer; dropout 0.1		
One layer LSTM	Unites 64		
Attention layer	Multiplicative attention; regularizer 1e-4		
Projection head	Dimension 256, 128, 64		
NT-Xent loss	Temperature parameter 0.1		
Pre-training	Epoch 100; batch size 512; SGD optimizer; cosine decay: decay steps 1000		
Linear evaluation	Epoch 300; optimizer: Adam learning rate 0.001; batch size 128;		
Fine-tuning	Epoch 200; optimizer: Adam learning rate 0.001; batch size 128		

**Table 7 sensors-24-00884-t007:** The performance of linear evaluation of different models with different training set proportions on the Aruba-1 dataset.

Label Fraction	SimCLR [24] Linear Evaluation		SimCLR + SAM Linear Evaluation		AttSimCLR Linear Evaluation		AttCLCHAR Linear Evaluation	
	F1 Score (%)	Accuracy (%)	F1 Score (%)	Accuracy (%)	F1 Score (%)	Accuracy (%)	F1 Score (%)	Accuracy (%)
1%	83.40	83.92	80.67	80.87	82.60	82.89	82.99	83.13
5%	84.88	84.98	82.53	82.63	84.21	84.36	84.63	84.59
10%	85.05	85.08	82.67	82.66	84.71	84.73	85.09	85.05
20%	85.11	85.11	83.06	83.05	85.08	85.13	85.41	85.36
30%	85.17	85.18	83.08	83.16	85.16	85.11	85.68	85.63

**Table 8 sensors-24-00884-t008:** The performance of linear evaluation of different models with different training set proportions on the Milan dataset.

Label Fraction	SimCLR [24] Linear Evaluation		SimCLR + SAM Linear Evaluation		AttSimCLR Linear Evaluation		AttCLCHAR Linear Evaluation	
	F1 Score (%)	Accuracy (%)	F1 Score (%)	Accuracy (%)	F1 Score (%)	Accuracy (%)	F1 Score (%)	Accuracy (%)
10%	59.74	63.42	61.52	63.93	61.10	63.43	61.28	63.40
20%	60.90	63.67	61.63	63.96	61.01	63.36	61.46	63.52
30%	61.53	64.01	61.89	64.01	61.32	63.60	61.63	63.94
40%	61.74	64.25	62.06	63.96	61.42	63.54	61.94	64.08
50%	62.09	64.39	62.36	64.25	61.33	63.53	62.05	64.17

**Table 9 sensors-24-00884-t009:** The performance of fine-tuning of different models with different training set proportions on the Aruba-1 dataset.

Label Fraction	SimCLR [24] Fine-Tuning		SimCLR + SAM Fine-Tuning		AttSimCLR Fine-Tuning		AttCLCHAR Fine-Tuning	
	F1 Score (%)	Accuracy (%)	F1 Score (%)	Accuracy (%)	F1 Score (%)	Accuracy (%)	F1 Score (%)	Accuracy (%)
1%	89.16	89.12	88.20	88.08	88.56	88.52	88.72	88.60
5%	93.70	93.67	93.59	93.57	93.65	93.63	93.85	93.82
10%	95.37	95.36	95.44	95.42	95.28	95.27	95.60	95.59
20%	96.81	96.80	96.79	96.77	97.00	96.99	97.04	97.02
30%	97.51	97.50	97.27	97.25	97.57	97.56	97.64	97.64

**Table 10 sensors-24-00884-t010:** Evaluation metrics for the semi-supervised learning scenario for the Aruba-1 dataset.

Model	Type	Accuracy (%)	Precision (%)	Recall (%)	F1 Score (%)
PNN [30]	Sup.	90	74	80	74
DNN [58]	Sup.	93	91	90	90
Bi-LSTM [59]	Sup.	98.15	91.7	91.7	91.7
Transformer [57]	SSL	-	95.9	96.9	96.4
AttCLCHAR (10%)	SSL	95.59	95.66	95.54	95.60
AttCLCHAR (20%)	SSL	97.02	97.07	97.00	97.04

**Table 11 sensors-24-00884-t011:** The performance of fine-tuning of different models with different training set proportions on the Milan dataset.

Label Fraction	SimCLR [24] Fine-Tuning		SimCLR + SAM Fine-Tuning		AttSimCLR Fine-Tuning		AttCLCHAR Fine-Tuning	
	**F1 Score (%)**	**Accuracy (%)**	**F1 Score (%)**	**Accuracy (%)**	**F1 Score (%)**	**Accuracy (%)**	**F1 Score (%)**	**Accuracy (%)**
10%	82.62	82.58	81.01	80.95	81.86	81.82	81.36	81.30
20%	90.20	90.17	88.97	88.92	90.52	90.46	90.17	90.11
30%	93.33	93.29	92.30	92.26	93.62	93.60	93.41	93.37
40%	94.61	94.60	94.28	94.27	95.07	95.03	95.21	95.19
50%	95.31	95.30	94.87	94.87	95.70	95.69	95.75	95.73

**Table 12 sensors-24-00884-t012:** Evaluation metrics for the semi-supervised learning scenario for the Milan dataset.

Model	Type	Accuracy (%)	Precision (%)	Recall (%)	F1 Score (%)
PNN [30]	Sup.	80	66	64	64
HAR_WCNN [60]	Sup.	95.35	85.90	87.22	86.43
LSTM [11]	Sup.	93.42	93.67	93.67	93.33
Ens2-LSTM [11]	Sup.	94.24	94.33	94.33	94.00
AttCLCHAR (30%)	SSL	93.37	93.57	93.25	93.41
AttCLCHAR (40%)	SSL	95.19	95.31	95.10	95.21

**Table 13 sensors-24-00884-t013:** The performance of linear evaluation and fine-tuning with different training set proportions on the Aruba-1 dataset.

Proportion	SimCLR [24] Linear Evaluation		AttCLCHAR Linear Evaluation		SimCLR [24] Fine-Tuning		AttCLCHAR Fine-Tuning	
	F1 Score (%)	Accuracy (%)	F1 Score (%)	Accuracy (%)	F1 Score (%)	Accuracy (%)	F1 Score (%)	Accuracy (%)
1%	84.13	84.47	82.76	82.83	89.17	89.13	88.98	88.93
5%	85.31	85.38	84.77	84.77	93.68	93.66	94.13	94.13
10%	85.42	85.43	84.95	84.93	95.28	95.27	95.58	95.56
20%	85.43	85.38	85.19	85.27	96.87	96.86	97.06	97.05
30%	85.47	85.42	85.22	85.15	97.49	97.48	97.54	97.53

## Data Availability

Data can be made available to interested readers by contacting the authors.

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
