# Peer review of "Enhancing Human Activity Recognition in Smart Homes with Self-Supervised Learning and Self-Attentionâ€"

_sensors, 2024, doi:10.3390/s24030884_

Round 1
Reviewer 1 Report
Comments and Suggestions for Authors
Authors have written with good readability.
All sections have sufficient information to understand.
Following are the major concerns:
All figures should be at least of 300 dpi resolution.
The proposed architecture, is complex. How it is efficient in terms of model size and training & testing time? Analysis should be shown in result section.
Reviewer 2 Report
Comments and Suggestions for Authors
This manuscript presentes a novel AttCLCHAR model based on the SimCLR framework with Self-Attention and SAM for HAR using ambient sensor data from smart homes. The model combines CNN, LSTM, and Self-Attention networks in the encoder to capture local and global features in time-series data. Based on the extensive experiment results on three publicly available smart home datasets, the proposed model has more robust performance in semi-supervised learning and transfer learning, compared with three existed models including SimCLR, SimCLR with SAM and AttSimCLR.
There are several comments and suggestions that should be addressed.
1. If it is feasible, please add the figures of time series data corresponding to different ambient sensors and different activities.
2. For different types of sensors, are the methods of feature extraction different? The methods of feature extraction should be concluded briefly in this paper.
3. In Table 8, the minimum value of F1 score is highlighted in bold for SimCLR, while the maximum value of Fi score is highlighted for SimCLR+SAM. Why the marking methods are different? Please use the same marking method for all Tables.
4. Please add the comparison results of training process, such as recognition accuracy curves, loss curves, feature scatter diagrams and cofusion matrixes.
Reviewer 3 Report
Comments and Suggestions for Authors
The article proposes an approach, AttCLCHAR, for Human Activity Recognition (HAR) in smart homes. Using the SimCLR framework with Self-Attention and SAM, the model integrates CNN, LSTM, and Self-Attention networks in the encoder to obtain temporal features comprehensively. The research conducts experiments on three publicly available smart home datasets, comparing AttCLCHAR with SimCLR, SimCLR with SAM, and AttSimCLR.
Positive aspects:
1) The integration of Self-Attention and SAM with SimCLR presents an approach to HAR that shows a thoughtful combination of established techniques.
2) AttCLCHAR demonstrates robustness in semi-supervised learning and transfer learning scenarios, outperforming other models on multiple datasets.
3) The evaluation on three different smart home datasets adds credibility to the results and illustrates the adaptability of the model to different environments.
4) The article is well structured, with a clear presentation of the model architecture, experimental setup, and detailed results, facilitating understanding.
Negative aspects:
1) The article acknowledges the challenges associated with unbalanced activities, but lacks specific strategies or experiments to address this issue. Future work should explore solutions to ensure balanced performance across activities.
2) While transfer learning results are presented, a more in-depth analysis of the impact of pre-training on Aruba-2 and its effectiveness in different downstream tasks would strengthen the conclusions.
3) The article briefly discusses the practical applications and potential challenges of implementing AttCLCHAR in real-world smart home environments. Presenting an extension on the feasibility of real-world applications would have strengthened the impact of the research.
Language and style:
The overall language of the article is clear and well-structured, with minimal grammatical errors. However, some sentences could benefit from more concise phrasing to improve readability. The terminology used is appropriate for the audience.
Conclusion:
The article presents a model, AttCLCHAR, for HAR in smart homes and demonstrates its performance in semi-supervised learning and transfer learning scenarios. While the model design and evaluation are highly, addressing unbalanced activities and providing more insight into real-world deployment challenges would strengthen the study. Given these points, the article has the potential for acceptance with minor revisions. The authors are encouraged to refine the aspects discussed to improve the practical relevance and completeness of the research.
Comments on the Quality of English LanguageMinor editing of English language required.
